# Circulating cell-free DNA from plasma undergoes less fragmentation during bisulfite treatment than genomic DNA due to low molecular weight

Bonnita Werner[1], Nicole Laurencia Yuwono[1], Claire Henry[1], Kate Gunther[1], Robert William Rapkins[1], Caroline Elizabeth Ford[1], Kristina Warton[1,2]*

1 School of Women's and Children's Health, University of New South Wales, Kensington, Sydney, Australia,
2 Genomics and Epigenetics Division, Garvan Institute of Medical Research, Darlinghurst, Sydney, Australia

* k.warton@unsw.edu.au

**Data Availability Statement:** All relevant data are within the paper.

## Abstract

### Background

Methylation patterns in circulating cell-free DNA are potential biomarkers for cancer and other pathologies. Currently, bisulfite treatment underpins most DNA methylation analysis methods, however, it is known to fragment DNA. Circulating DNA is already short, and further fragmentation during bisulfite treatment is of concern, as it would potentially reduce the sensitivity of downstream assays.

### Methods

We used high molecular weight genomic DNA to compare fragmentation and recovery following bisulfite treatment with 2 commercially available kits (Qiagen). The bisulfite treated DNA was visualised on an agarose gel and quantified by qPCR. We also bisulfite treated, visualised and quantitated circulating DNA from plasma.

### Results

There was no difference in DNA fragmentation between the two kits tested, however, the Epitect Fast kit gave better recovery than the standard Epitect kit, with the same conversion efficiency. We also found that bisulfite treated circulating DNA migrates as distinct bands on agarose gels, suggesting that, in contrast to genomic DNA, it remains largely intact following treatment. Bisulfite treatment of 129 and 234 base PCR products confirmed that this was due to the short length of the circulating DNA fragments. Compared to double stranded DNA, bisulfite treated single stranded DNA gives a very weak signal on gel electrophoresis.

### Conclusions

DNA fragmentation during bisulfite treatment does not contribute to loss of sensitivity in methylation analysis of circulating DNA. The absence of DNA fragments below

**Funding:** No authors were paid a salary directly via an external funding body, however, the salary of KW is from an Ovarian Cancer Research Foundation Grant, CH salary is funded by Gynaecological Oncology Fund, and RR salary is funded by Cure Brain Cancer foundation, with all salaries administered by the University of New South Wales. NY is funded by an Australian Government Research Training Program Scholarship with a Translation Cancer Research Network top up scholarship. RR is funded by. CF is funded as an employee of the University of New South Wales. The funders had no role in study design, data collection and analysis, decision to publish or preparation of the manuscript.

**Competing interests:** I have read the journals competing interests policy and the authors of this manuscript have the following competing interests: KW- stock ownership Guardant Health, Exact Sciences and Epigenomics AG. This does not alter our adherence to PLOS ONE policies of sharing data and materials.

approximately 170 bases from agarose gel images of purified circulating DNA raises the possibility that these fragments are single stranded following the DNA extraction step.

## Introduction

Methylation analysis of circulating cell-free DNA (cirDNA) in blood plasma offers scope for the identification of cancer biomarkers[1, 2], as well as determining the tissues types that contribute to the cirDNA pool[3, 4]. DNA methylation is particularly relevant in the field of cancer diagnostics, since it is more consistent between individual tumours than mutation, and thus enables PCR detection of tumour DNA without *a priori* knowledge of the tumour mutation profile[1].

Since it was first described in 1992[5], bisulfite treatment has been the mainstay of DNA methylation analysis. Bisulfite reacts with unmethylated cytosine, resulting in conversion to uracil, while methylated cytosine is reacts at a much lower rate and the majority of residues remain unchanged. Thus, the cytosine methylation status of a DNA region can be determined by comparison of the sequence before and after bisulfite treatment.

Bisulfite treatment is known to fragment DNA as a side effect of the low pH and high temperature required for complete conversion of unmethylated cytosine to uracil. When working with high molecular weight genomic DNA, the fragmentation contributes to full conversion of the sample as it creates DNA pieces within a size range that is readily denatured, that is, no larger than 2 kb[6]. However, DNA fragmentation has been of concern when working with cirDNA, because cirDNA is already highly fragmented, with most molecules occurring at a size of 167 bases, presumably reflecting its apoptotic origin[7]. The small size of cirDNA makes it difficult to both purify[8, 9], and to detect by PCR, as any DNA molecules that contain breaks within the PCR target sequence are not available for amplification[10]. Hence, exploiting DNA methylation for cancer biomarker development has been regarded as a trade-off between the high consistency of tumour methylation patterns and the decreased assay sensitivity due to fragmentation of DNA targets by bisulfite treatment.

In this study, we visualised bisulfite treated cirDNA on an agarose gel, and showed that it undergoes relatively little fragmentation compared to high molecular weight genomic DNA. We also showed that this relative stability under bisulfite treatment conditions is due to the small size of the cirDNA fragments. This is the first time that this property of cirDNA has been demonstrated and it has implications for the development of cancer detection tests and liquid biopsies that target the low molecular weight fraction of cirDNA. We also compared two commercially available bisulfite conversion kits for DNA recovery and fragmentation.

## Materials and methods

### Biospecimens

This study was approved by the University of New South Wales Human Research Ethics Committee, approval number HC17020. All participants gave written informed consent. Blood samples from healthy female donors with an age range of 21 to 47 years were used. Blood was drawn into 10 mL EDTA tubes (Becton Dickenson), and plasma was separated by centrifugation at 2500 xg at 4˚C for 10 minutes, followed by a second spin of the plasma at 3500 xg at 4˚C for 10 minutes. All blood samples were processed into plasma within 3 hours of collection and plasma was stored at -80˚C until use. All plasma samples were used in experiments within 8 months of the blood collection procedure. Plasma was thawed at room temperature, and any

leftover plasma unused after thawing was discarded, i.e. each plasma sample had only been frozen once. Human genomic DNA (gDNA) purchased from Roche (Cat #11691112001) and stored at 4˚C as specified by the manufacturer was used as the high molecular weight DNA sample.

## CirDNA extraction

cirDNA was extracted from the plasma samples using the QIAamp Circulating Nucleic Acid (CNA) kit (cat# 55114, Qiagen) according to manufacturer's instructions, however, for gel visualisation of cirDNA, we increased the plasma input volume in order to obtain a highly concentrated sample. We have previously shown that the CNA kit protocol can be scaled up to accommodate up to 17.5 mL plasma input with no loss of purification efficiency[11]. In this study, to visualise bisulfite treated cirDNA on a gel, a total of 102 mL of plasma from 5 donors was pooled and processed in 6 extractions of 17 mL plasma, with appropriate scaling of reagents up until the elution step. At the elution step, a single 25 µL aliquot of Elution Buffer was passed over all 6 columns, followed by a second aliquot of 25 µL then a third aliquot of 25 µL. This resulted in the cirDNA from 102 mL of plasma being collected in a total elution volume of approximately 60 µL, taking into account buffer losses on the columns.

For comparison of yield from the Epitect standard and the Epitect Fast kit, cirDNA was extracted from a total of 20 mL of plasma in 4 x 5 mL extractions with 84 µL elution volumes for each extraction. The eluted DNA was pooled to obtain a total volume of ~320 µL.

## Bisulfite conversion of gDNA

1.5 µg of gDNA were bisulfite treated using either the Epitect kit (Qiagen, Cat# 59104, here referred to as Epitect standard kit) or the Epitect Fast kit (Qiagen, Cat# 59824). When using the Epitect Fast kit, the 60˚C incubation steps were carried out for either 10 minutes or 20 minutes, as indicated in the figure legends. No carrier RNA was added to the reactions. All bisulfite treated gDNA samples were eluted in 25 µL of Elution Buffer.

To assess the efficiency of the bisulfite conversion, samples underwent amplification of the cancer associated gene, *MGMT* using the MGMT Pyro Kit (Qiagen). The kit detects methylation levels for five CpG sites spanning exon 1 with a cytosine not associated with a CpG site serving as an internal control for bisulfite conversion.

## Bisulfite conversion of cirDNA

For comparison of yield from Epitect standard and Epitect Fast kit, cirDNA extracted from a total of 20 mL of blood plasma and pooled in ~320 µL of Elution Buffer was treated in two separate experiments in duplicate 40 µL volumes with each of the 2 protocols (total of 4 replicates per protocol). The 60˚C incubations of the Epitect fast protocol were carried out for 20 minutes. No carrier RNA was added to the reactions. Both the Epitect standard and the Epitect Fast bisulfite treatments had a 25 µL elution volume.

For gel visualisation, cirDNA extracted from 102 mL of plasma was bisulfite converted using the Epitect Fast kit, with 40 µL of purified cirDNA converted per reaction. No carrier RNA was added to the reactions. The cirDNA was eluted in 33 µL of Elution Buffer.

## Bisulfite conversion of PCR products

For each PCR product, 40 µL from a total 50 µL of PCR reaction volume were bisulfite converted using the Epitect Fast kit and 20 minute 60˚C incubation times. Converted DNA was eluted in 25 µL of Elution Buffer.

## Quantitative PCR

All PCR reactions were carried out on a Biorad CFX96 Real Time PCR machine. Bisulfite converted DNA was measured by quantitative PCR (qPCR) of the *GSTP1* (NM_000852.3) gene promoter using methylation non-specific primers (forward primer: `TTTGTGAAGIGGGT GTGTAA`; reverse primer: `CAAATCCCCAACIAAACCTA`; product size 148 bases). PCR reactions were set up to contain 1 x buffer, 0.2 mM each dNTP (New England Biolabs, Cat# N0447S), 3 mM MgCl₂, 0.2 μM of each primer, 1/10 000 dilution of SYTO9 (Thermo Fisher Scientific, Cat# S34854) and 0.16 μL per reaction of Platinum Taq Polymerase (Invitrogen, Cat# 10966). Cycling was 95˚C for 3 minutes, then 95˚C for 5 seconds, 58˚C for 20 seconds and 72˚C for 30 seconds for 45 cycles, followed by a melt curve to confirm reaction specificity. A no template control with water replacing the DNA template was included in all PCR reactions, and no amplification was observed.

A standard curve made with Epitect standard kit converted DNA was used to obtain relative quantitation of genomic DNA and cirDNA. Samples containing DNA equivalent to 1.2 ng genomic DNA input converted using the two different kits were quantified against the standard curve. A total of 4 replicates from 2 separate bisulfite conversion experiments (total of 4 replicates of each protocol) was quantitated by qPCR in triplicate.

Bisulfite treated cirDNA from quadruplicate replicates of the Epitect standard and Epitect Fast kit was quantitated in triplicate by qPCR of the *GSTP1* promoter against the standard curve described above.

## PCR for bisulfite treatment of short DNA fragments

DNA fragments of 234 and 129 bases were generated by PCR of *SNAI1* (NM_0059850) (F primer `CCTCCCTGTCAGATGAGGAC`; R primer `CCAGGCTGAGGTATTCCTTG`) and *IDH1* (NM_005896) (F primer: `CGGTCTTCAGAGAAGCCATT`; R primer `GCAAATCACA TTATTGCCAAC`) respectively. PCR reaction composition was as for *GSTP1* above, scaled up to 50 μL reaction volume, with cycling of 95˚C for 3 minutes, then 95˚C for 10 seconds, 60˚C for 20 seconds and 72˚C for 15 seconds for 46 cycles. From each 50 μL PCR reaction, 4 μL of untreated PCR product was loaded on a gel, while 40 μL was bisulfite converted as described below.

## Gel electrophoresis

Agarose gels were made up with 1% agarose (Lonza Seakem, Cat# 50002) in 40 mM Tris, 20 mM acetic acid, 1 mM EDTA, pH 8.4 buffer (TAE buffer), plus 10 μL Gel Red stain (Biotium, Cat# 41003) per 100 mL added just before the gel was poured. For gels visualising cirDNA, 0.1 μL of 100 bp DNA ladder MWM (molecular weight markers) (New England Biolabs, Cat# N3231S) was loaded, alongside the sample volumes indicated in the figure. For gels visualising PCR product 1 μL of DNA ladder MWM was loaded alongside 4 μL of control untreated PCR product and 25 μL of bis-treated PCR product, that had been obtained from 40 μl of untreated PCR product. Gels were run at 100 V for the times indicated in the figure legends.

## Statistical analysis

p-values for DNA quantitation data were calculated using unpaired one-tailed t-tests with Graphpad Prism 8 software.

## Results

### Epitect Fast kit versus Epitect standard kit bisulfite conversion of genomic DNA and cirDNA

Epitect Fast kit was compared against the standard Epitect kit for recovery and size of genomic DNA after bisulfite conversion (Fig 1A and 1B). The Epitect Fast kit provides the option of carrying out the 60˚C incubation steps for either 10 minutes or 20 minutes, and we compared both using genomic DNA. We used qPCR and a standard curve approach to obtain relative quantitation of Epitect standard and Epitect Fast bisulfite converted DNA. The standard curve was constructed using genomic DNA converted with the Epitect standard kit and used to quantitate bis-treated samples containing DNA equivalent to 1.2 ng genomic DNA. As expected, the sample containing Epitect standard-treated DNA returned an apparent DNA quantity of approximately 1.2 ng. The sample containing Epitect Fast-treated DNA returned an apparent DNA quantity of approximately 2.6 ng using both the 10 minute and the 20 minute conversion protocol. For the purpose of calculating the relative differences in yield shown in Fig 1B, the Epitect standard samples were set to '1'. Epitect Fast kit was found to give around 2-fold higher recovery. Our results on DNA recovery are similar to those of Holmes and colleagues, who reported a higher or similar recovery with the Epitect Fast kit than the Epitect standard kit, however, this was dependent on the PCR target used for quantition[12].

We also observed a higher recovery of cirDNA treated with the Epitect Fast kit, with the apparent cirDNA concentration in plasma of 0.44 ng/mL when using the Epitect standard kit and 1.02 ng/mL when using the Epitect Fast kit (Fig 1C).

We did not observe any decrease in the fragmentation of genomic DNA using the Epitect Fast protocol (Fig 1A). Both protocols resulted in DNA that was highly fragmented compared to the input material, with most of the sample appearing at below 1200 bases on an agarose gel. This is in contrast to the work of Holmes and colleagues, who showed a small decrease in fragmentation when using the Epitect Fast/FFPE bisulfite kit, however, the difference was very slight[12]. G-C rich sequences that contain unmethylatated cytosines are known to undergo more fragmentation than A-T rich sequences during bisulfite treatment[13], so individual loci may respond differently to conversion protocols, but we observed no difference in overall fragmentation between the two methods.

Genomic DNA samples converted using the Epitect Fast kit and the Epitect standard kit were checked for completion of bisulfite conversion using the MGMT Pyro Kit. The assay assesses 5 CpG sites within exon 1 of the tumour suppressor gene *MGMT*, and interrogates a cytosine nucleotide not followed by a guanine in exon 1 to determine bisulfite conversion. Both methods were found to give ≥97% bisulfite conversion. This is very similar to the results of Holmes and colleagues, who reported a 98.7% and 99.8% conversion efficiency as determined by clonal Sanger sequencing when using the Epitect standard and Epitect Fast kits respectively[12]. Based on the higher yield, Epitect Fast kit was chosen for treatment of the cirDNA samples for gel visualisation.

### CirDNA undergoes less fragmentation than genomic DNA during bisulfite treatment

In pilot experiments, we consistently found that while untreated cirDNA is readily visualisable on a gel, bisulfite treated cirDNA does not produce a visible band. We tried a number of approaches to overcome this. Initially, we incubated the gel on ice prior to UV viewing, based on the premise that the bis-treated cirDNA is not visible because it is single stranded, and thus has a low affinity for Gel Red dye. Ice bath incubation, which should have increased the amount of double stranded DNA, did not result in visible bis-treated cirDNA.

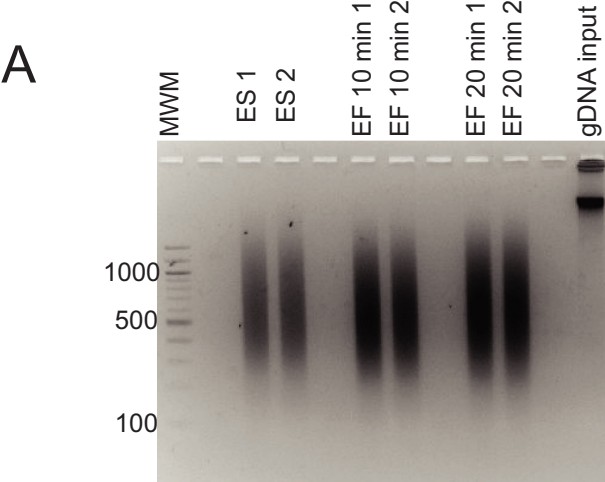

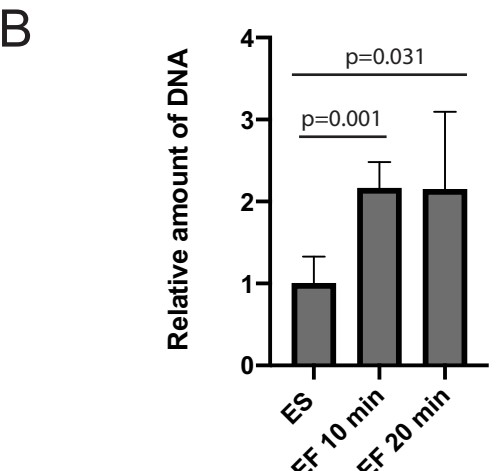

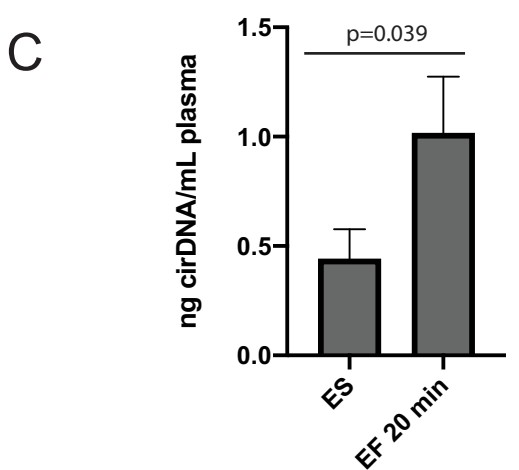

**Fig 1.** (A) Agarose gel of gDNA bisulfite converted using either the ES or the EF kit. (B) Relative quantification of gDNA recovery following either ES or EF kit conversion. (C) cirDNA recovery following ES or EF kit conversion. 60˚C incubation times for the EF kit are as indicated. B and C technical replicates n = 4; PCR replicates n = 3. ES–Epitect Standard kit; EF–Epitect Fast kit.

We then increased the starting plasma input volume, such that the amount of bis-treated material loaded per lane was the equivalent to cirDNA extracted from 20 mL of blood plasma. This also did not result in visible bis-treated DNA.

Finally, we increased the bis-treated cirDNA to the entire amount extracted from 55 mL of plasma and viewed the agarose gel under UV light after only 15 minutes of electrophoresis. With this approach, we were easily able to visualise the bis-treated cirDNA (Fig 2). With 15 minutes of electrophoresis, cirDNA from both 55 mL and 20 mL of plasma was sufficient to produce strong bands.

Several aspects of the data are notable. Firstly, distinct cirDNA bands are visible, suggesting that a large proportion of the DNA is not fragmented by bisulfite treatment, as fragmentation would likely have generated random sized DNA lengths that migrate as a smear. The cirDNA result is in contrast to genomic DNA, where all the bis-treated DNA migrated as a broad smear, between approximately 300 and 1200 bases, with no distinct bands present (Fig 1A). The bis-treated cirDNA bands migrated somewhat below the corresponding untreated cirDNA sample, with the difference most likely reflecting not change in length, but faster migration of single stranded DNA compared to double stranded DNA.

Secondly, the bis-treated DNA is largely lost from the gel image following 40 minutes of electrophoresis. Fig 2A shows that after 15 minutes, the band of bis-treated cirDNA extracted from 55 mL of plasma is much stronger than the control cirDNA extracted from 5 mL of plasma. The bis-treated cirDNA extracted from 20 mL of plasma also gives a very strong signal. In contrast, after 40 minutes (Fig 2B), the bis-treated cirDNA band from 55 mL is much weaker than the control cirDNA band, and the bis-treated cirDNA band from 20 mL is almost invisible.

We postulated that the presence of discrete bands in bis-treated cirDNA samples was due to short DNA pieces being less susceptible to fragmentation during bisulfite treatment than long DNA pieces, and tested this by bisulfite treating PCR products of 243 and 129 bases. We found that these PCR products also underwent relatively little fragmentation, migrating as discrete bands after bisulfite treatments (Fig 3). The amount of bis-treated PCR product loaded on the gel was equivalent to 10-fold the amount of the corresponding untreated PCR product loaded, but resulted in relatively weak bands, highlighting the difficulties of visualising single-stranded DNA. The more rapid loss of the bis-treated PCR product from the gel compared to the control PCR product was confirmed by quantitation of the relative band intensities (S1 Fig).

## Discussion

Bisulfite treatment is well known to fragment DNA, however, we found that cirDNA persisted as discrete bands on an agarose gel after bisulfite conversion. This is due to greater stability of short DNA fragments, as demonstrated by bisulfite treatment of 234 and 129 bp PCR products. These data suggest that methylation biomarkers in cirDNA are not likely to suffer a decrease in sensitivity due to target fragmentation during bisulfite treatment. High molecular weight DNA, within the size range shown in Fig 1A and 1B, is only rarely observed in plasma samples [7], but when present will become fragmented after bisulfite treatment. This caveat applies in particular to blood samples from patients with cancer and other severe diseases, in whom higher molecular weight cirDNA and greater inter-individual variation are more likely.

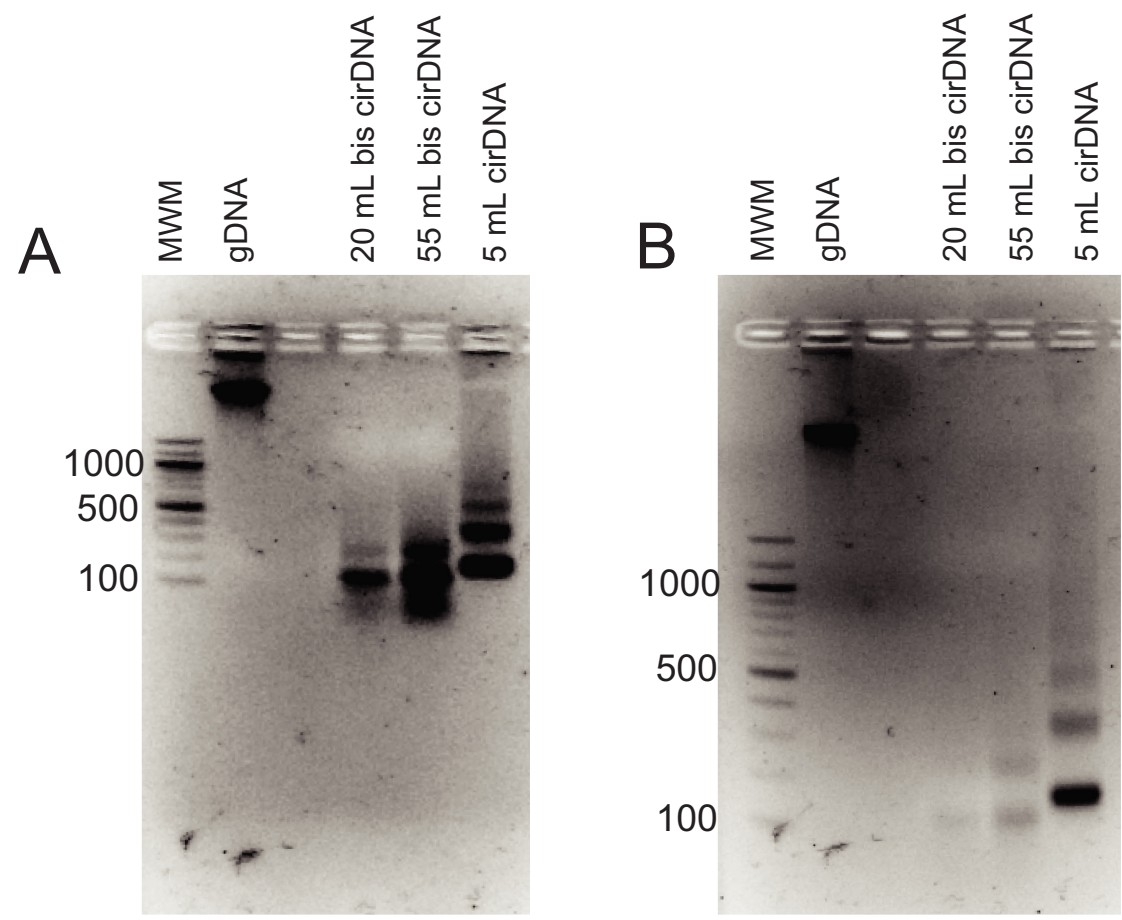

**Fig 2.** Agarose gel of cirDNA and bis-treated cirDNA following (A) 15 minutes and (B) 40 minutes of electrophoresis. The amount of cirDNA in each lane corresponds to the total cirDNA extracted from the plasma volumes indicated. gDNA–untreated genomic DNA.

We also found that the bis-treated DNA becomes less visible on the gel with increasing electrophoresis time. There are a number of mechanisms that might explain this. One is that single stranded DNA is more susceptible to DNAses than double stranded DNA and is progressively cleaved as the gel is run. The agarose gel running buffer does contain EDTA, which is generally inhibitory for DNAses, but it is possible that some DNAse activity remains. Another potential reason is that the Gel Red dye migrates towards the cathode, thus once the DNA enters the anode-most part of the gel, it is in an area where the dye concentration is insufficient to visualise single stranded DNA. However, we had previously attempted to post-stain gels to visualise bis-treated cirDNA, and this had not been effective, suggesting that dye concentration is not the main factor. A third mechanism is that the bis-treated DNA, which is single stranded, forms intramolecular and intermolecular secondary structures during migration, resulting in a sample that becomes less and less uniform with increasing electrophoresis time, and thus no longer migrates as a discrete band. If this is the case, it would suggest that the smearing visible around the bis-treated cirDNA bands is due to a range of secondary structures being present, rather than DNA fragmentation. Finally, DNAse digestion, decreased dye concentration and DNA secondary structure may all independently contribute to bis-treated cirDNA being difficult to visualise.

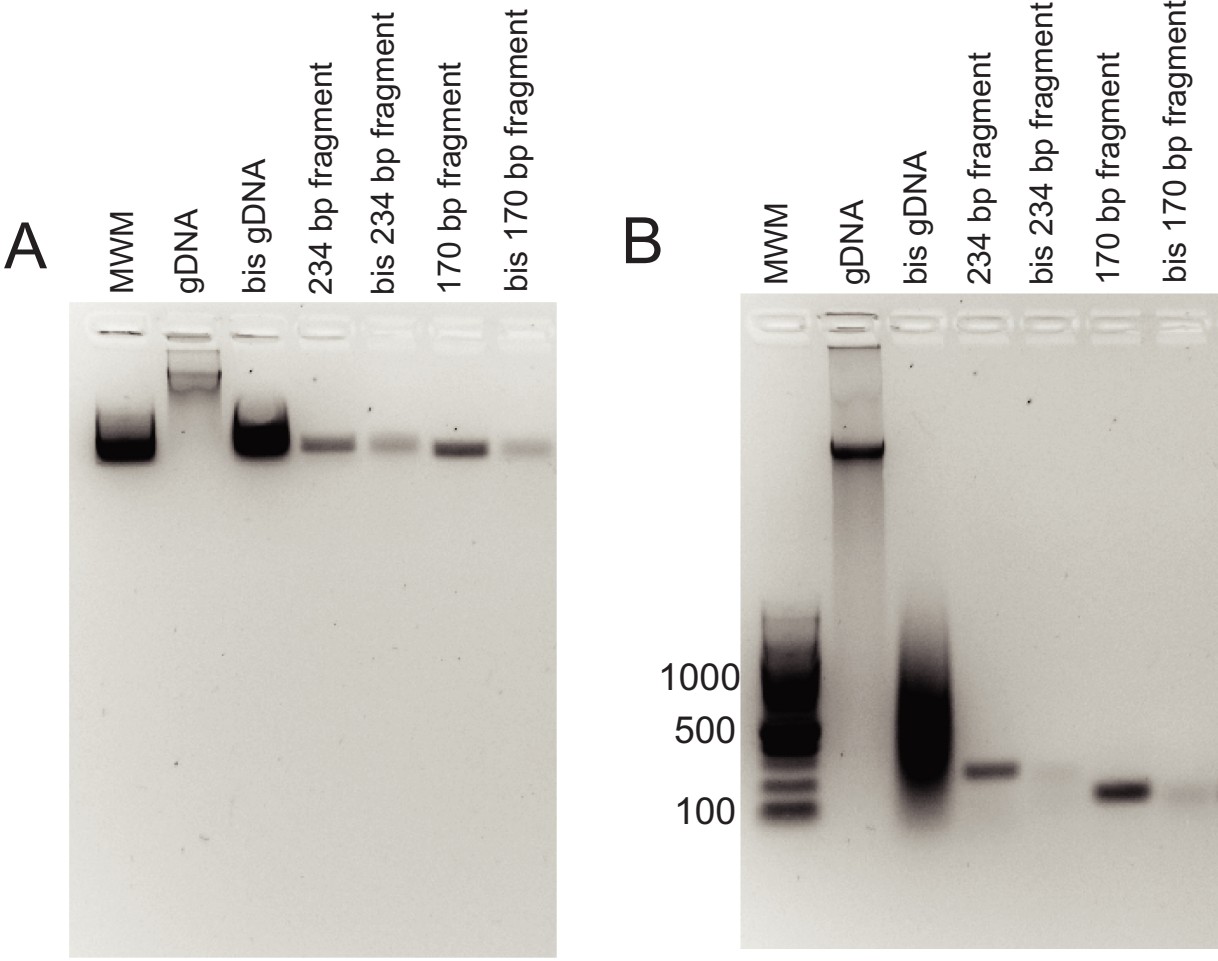

**Fig 3.** Agarose gel of PCR product and bis-treated PCR product following (A) 15 minutes and (B) 40 minutes of electrophoresis. Size of each PCR product is as indicated. gDNA–untreated genomic DNA; bis gDNA–bis-treated genomic DNA.

Our results also highlight a discrepancy between cirDNA whole genome sequencing data and cirDNA agarose gel appearance. It has been shown that single-stranded, but not double-stranded, sequencing library preparations identify a substantial fraction of cirDNA that is below 167 bases in size[14]. This was assumed to be because the protocols are better at capturing short DNA molecules, as they exclude a size restriction step[14]. However, even when a large amount of cirDNA is run on an agarose gel (Fig 2A), no DNA migrating below the main 167 base band is visible. This raises the possibility that the short DNA molecules are not only short, but also single stranded. Thus, they would be difficult to visualise on agarose gels for the same reasons, described above, that single-stranded bis-treated DNA is difficult to visualise.

If this is the case, the question arises of whether DNA fragments below 167 bases are endogenously single-stranded in blood plasma, or whether they become single-stranded as a side-effect of the DNA purification process. We believe the latter to be more likely. A commonly used method of cirDNA extraction, and the one used by us in this study, is the Circulating Nucleic Acids kit from Qiagen. This kit protocol includes a plasma proteinase K digestion step which is carried out for 30 minutes at 60˚C in high concentration guanidine salt. It is likely that under these conditions, short DNA fragments become denatured. With this in mind, we ran side-by-side plasma cirDNA extractions using a 40˚C digest step as well as the standard

60˚C digest step. The decreased temperature resulted in a slight decrease in overall yield, but did not produce short cirDNA fragments that were visible on an agarose gel (data not shown). The reason why cirDNA fragments below 167 bases are apparent in single-stranded cirDNA sequencing libraries, but can't be seen on a gel, remains unclear.

In conclusion, we have used agarose gel electrophoresis to show that DNA fragments below 234 bases, including cirDNA, undergo relatively little fragmentation during bisulfite treatment, and that single-stranded DNA is difficult to visualise on an agarose gel. The stability of short DNA fragments is significant for efforts to utilise methylation as a diagnostic target for cirDNA assays, since it shows that additional DNA fragmentation due to bisulfite treatment is not likely to impact assay sensitivity. High sensitivity is a critical consideration when developing cancer diagnostic and monitoring assays based on cirDNA.

## Supporting information

**S1 Fig. Quantitation of ssDNA on agarose gel during electrophoresis timecourse.** Agarose gel ratio of control to bisulphite treated PCR product band intensity after 15 minutes and 40 minutes of electrophoresis for (A) SNAI1 and (B) IDH1.
(AI)

**S2 Fig. Raw gels.**
(PDF)

## Author Contributions

**Conceptualization:** Caroline Elizabeth Ford, Kristina Warton.

**Investigation:** Bonnita Werner, Nicole Laurencia Yuwono, Claire Henry, Kate Gunther, Robert William Rapkins, Kristina Warton.

**Methodology:** Claire Henry, Kristina Warton.

**Supervision:** Kristina Warton.

**Writing – original draft:** Kristina Warton.

**Writing – review & editing:** Bonnita Werner, Nicole Laurencia Yuwono, Claire Henry, Kate Gunther, Robert William Rapkins, Caroline Elizabeth Ford, Kristina Warton.

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
