## [Decision Letter · Decision Letter 0]

15 Aug 2019

PONE-D-19-20341

Circulating cell-free DNA from plasma undergoes less fragmentation during bisulphite treatment than genomic DNA due to low molecular weight

PLOS ONE

Dear Dr Warton,

Thank you for submitting your manuscript to PLOS ONE. After careful consideration, we feel that it has merit but does not fully meet PLOS ONE’s publication criteria as it currently stands. Therefore, we invite you to submit a revised version of the manuscript that addresses the points raised during the review process.

I have received the reviews of your manuscript. While your paper addresses an interesting question, the reviewers stated several concerns about your study where additional experimentation and documentation is needed.  Both reviewers voice a number of concerns regarding the used methodology for data analysis, and the limitations of the test system.  These comments need to be addressed carefully.  Please see reviewers' insightful comments below for detail.

We would appreciate receiving your revised manuscript by Sep 29 2019 11:59PM. To enhance the reproducibility of your results, we recommend that if applicable you deposit your laboratory protocols in protocols.io, where a protocol can be assigned its own identifier (DOI) such that it can be cited independently in the future. For instructions see: http://journals.plos.org/plosone/s/submission-guidelines#loc-laboratory-protocols

We look forward to receiving your revised manuscript.

Kind regards,

Baochuan Lin, Ph.D.

Academic Editor

PLOS ONE

a)    Please provide an amended Funding Statement that declares *all* the funding or sources of support received during this specific study (whether external or internal to your organization) as detailed online in our guide for authors at http://journals.plos.org/plosone/s/submit-now.  

b)    Please state what role the funders took in the study.  If any authors received a salary from any of your funders, please state which authors and which funder. If the funders had no role, please state: "The funders had no role in study design, data collection and analysis, decision to publish, or preparation of the manuscript."

"I have read the journal's policy and the authors of this manuscript have the following competing interests: KW - stock ownership in Guardant Health, Exact Sciences and Epigenomics AG. "

Reviewers' comments:

Reviewer's Responses to Questions

**Comments to the Author**

1. Is the manuscript technically sound, and do the data support the conclusions?

Reviewer #1: No

Reviewer #2: Partly

2. Has the statistical analysis been performed appropriately and rigorously? 

Reviewer #1: N/A

Reviewer #2: Yes

3. Have the authors made all data underlying the findings in their manuscript fully available?

Reviewer #1: Yes

Reviewer #2: Yes

4. Is the manuscript presented in an intelligible fashion and written in standard English?

Reviewer #1: Yes

Reviewer #2: No

5. Review Comments to the Author

Reviewer #1: Warton et al aim to demonstrate that cell free circulating DNA is less susceptible to fragmentation during bisulfite modification compared to genomic DNA.

The only way to assess this was visualisation on agarose gels using Gel Red Stain from Biotium.

Although these findings are generally interesting, further substantial work is required to support their conclusions:

Gel Red is an intercalating Nucleic Acid stain like Ethidium Bromide, and although less toxic, it will stain ssDNA less well despite the Biotium product description page claiming that Gel Red stains ssDNA; the aforementioned is even acknowledged on the Biotium website where it indicates that Gel Red stains ssDNA 50% less efficiently than dsDNA "Titration assays using a fluorescence microplate reader showed that the fluorescence signal of GelRed® bound to ssDNA and RNA is about half that of GelRed® bound to dsDNA." This may explain the issues of visualisation, and makes comparison with concentrations of dsDNA difficult. Alternative techniques are required to discriminate single from double-strand DNA.

In addition, it is mandatory to use alternative techniques to quantify DNA of different fragment sizes (i.e. the use of a Fragment Analyser).

The size of the cirDNA before and after bisulfite treatment should be reported (Fig 2), as the ambiguity with respect to size makes it very difficult to follow the comparison with the PCR fragment sizes used and specified in Fig 3 and the fragmentation of gDNA (300-1200 bp) (also Fig 3); the aforesaid clarification would help the discussion throughout the manuscript.

What is the nature of the “short DNA fragments” which are not found (line 67)? Do the authors mean the absence of fragmented bisulfite treated cirDNA which are smaller than the visualised sizes? It is important to accurately describe those sizes in the Results section, as “short” is relative and too vague in this context.

Reviewer #2: In their present study “Circulating cell-free DNA from plasma undergoes less fragmentation during bisulphite treatment than genomic DNA due to low molecular weight” Werner et al investigated the fragmentation of DNA caused by bisulfite reaction that is required for DNA methylation studies. The authors compared the fragmentation of high molecular weight DNA with fragmented apoptotic DNA in plasma. The manuscript is clear, concise, and well written. The results has some merit for a few researchers in the field. However, the study suffers from several limitations which should be addressed prior to its publication.

Major points

1. Test system: The authors used Human Genomic DNA from human blood (buffy coat) as a high molecular weight (>50kb according to the manufacturer) reference DNA sample. Such HMW DNA is usually not present in clinically relevant samples. Accordingly, the test system is of limited use. This should be critically discussed in the manuscript.

2. Novelty: Holmes et al. (PLoS One. 2014; 9(4): e93933.) have previously published that converting the same high molecular weight DNA using the same two Qiagen kits leads to >97% conversion; higher yields with the EpiTect Fast kit, and similar fragmentation patterns. Some of the results presented by Werner et al. are in perfect concordance with the results shown by Holmes et al.. Hence, the study by Holmes at al. should be appropriately included into the introduction and discussion of the paper.

3. Influence of extraction method on fragmentation: Both investigated kits use silica membrane spin column-based DNA purification after bisulfite conversion. This purification might have an influence on the fragmentation of HMW DNA irrespective of the bisulfite conversion. The authors should perform a control reaction in which they omit the bisulfite reagent and replace it with water in order to test the influence of the extraction method on the fragmentation.

4. Sequence-specific fragmentation: The analyzed only two loci (MGMT and GSTP1). Fragmentation of DNA due to bisulfite treatment might be sequence-specific. Do AT-rich sequences show differences regarding fragmentation as compared to GC-rich sequences which are obviously much more affected by the bisulfite reaction? This matter should be discussed.

5. In Figure 3A the authors showed that two PCR product showed the same band on the agarose gel before and after bisulfite conversion. The authors concluded that the fragmentation of the PCR product due to the bisulfite conversion is limited due to its small size and that the low concentration of the bisulfite converted PCR product is caused by the impaired visualization of ssDNA on an agarose gel. The results do not entirely support the conclusion. Since the authors used only 1% agarose gels, fragmented PCR products might not be visible. Other methods (e.g. 3.5% agarose gel, bioanalyzer, polyacrylamide gel) would be much better suited to visualize fragmented DNA. Secondly, the authors used PCR product as a surrogate DNA sample representing short DNA fragments. Even though the size is comparable to apoptotic DNA in plasma, PCR products might show a different behavior during the bisulfite conversion since the concentration of identical DNA sequences is much higher leading to a lower denaturation efficacy. It would be more convincing if the authors had compared the fragmentation of a large PCR product (~2kb) with their short ones (~200bp).

Minor points

1. Most scientist in the field use the spelling “bisulfite” instead of “bisulphite”. Using the more common spelling would increase the visibility of the article.

2. Since the nature of the bisulfite conversion is a major part of this study, its features should be described more accurately. It is incorrect that methylated cytosine are unreactive. Their conversion occurs at much lower rates compared to cytosines. It is not entirely correct that DNA degradation is caused by the high temperature alone. DNA degradation is mainly caused by depurination and depyrimidation leading to abasic sites, followed by DNA strand breaks due to N-glycoside bond cleavage. This depurination and depyrimidation is mainly due to the low pH needed for the bisulfite conversion. A high temperature additionally increases this problem.

3. Fig. 3: pb -> bp

4. Please use italicized gene names (IDH1, SNAI1, GSTP1).

5. The first paragraph extensively discusses features of gel electrophoresis regarding the visualization of small DNA fragments and in particular bisulfite converted small DNA fragments. Firstly, the scope of the study was not to describe the analytical performance of agarose gel electrophoresis, a method from the 1960s. Secondly, it is well known to people skilled in the art, that DNA disappears form an agarose gel after prolonged incubation, mainly due to diffusion. Thirdly, the authors used 1% agarose gel, a concentration that is not suited for small DNA fragments. In conclusion, the disappearance of small DNA fragments from the agarose gel after extended incubation time is most likely due to diffusion that is facilitated by the use of a too low concentrated agarose gel and prolonged incubation times. This paragraph of the discussion can be removed. In addition, Figures 2B and 3B can be removed since they contain basic knowledge that is already available to everyone in the field.

6. PLOS authors have the option to publish the peer review history of their article (what does this mean?). If published, this will include your full peer review and any attached files.

Reviewer #1: No

Reviewer #2: No

---

## [Author Response · Author response to Decision Letter 0]

3 Sep 2019

We thank the reviewers for their careful reading of our manuscript, and thoughtful feedback. Specific responses to their comments are listed below.

Reviewer 1

Comment

Gel Red is an intercalating Nucleic Acid stain like Ethidium Bromide, and although less toxic, it will stain ssDNA less well despite the Biotium product description page claiming that Gel Red stains ssDNA; the aforementioned is even acknowledged on the Biotium website where it indicates that Gel Red stains ssDNA 50% less efficiently than dsDNA "Titration assays using a fluorescence microplate reader showed that the fluorescence signal of GelRed® bound to ssDNA and RNA is about half that of GelRed® bound to dsDNA." This may explain the issues of visualisation, and makes comparison with concentrations of dsDNA difficult. Alternative techniques are required to discriminate single from double-strand DNA.

Response

Reviewer 1 is correct in pointing out that GelRed stains ssDNA less well than it does dsDNA. Unfortunately, there is currently no DNA stain that stains ssDNA efficiently, and this was one of the main challenges with visualising the cirDNA after bisulphite conversion. We acknowledge this limitation of the methodology by stating “the bis-treated cirDNA is not visible because it is single stranded, and thus has a low affinity for Gel Red dye” (lines 319-320). We attempted to modify the technique by cooling the gels, but without success. The value and novelty of our study is that we managed to visualise cirDNA following bisulphite treatment even in the absence of dyes that efficiently stain ssDNA. 

Comment

In addition, it is mandatory to use alternative techniques to quantify DNA of different fragment sizes (i.e. the use of a Fragment Analyser). 

Response

We did not attempt to run the samples on a Fragment Analyser-type platform for a number of reasons. Firstly, there is not published or anecdotal data that the dyes used in this type of system have any better affinity for ssDNA than GelRed. Secondly, we found that visualising the DNA depends on a large sample input volume, and this is not feasible with a BioAnalyser/Tape Station. Finally, we had successfully visualised the bis-converted cirDNA on an agarose gel, so did not proceed to other methods. We do however appreciate the suggestion, and will do this in the future, as this type of electrophoresis system would avoid DNA diffusion out of the gel, and may provide a stronger signal. 

Comment

The size of the cirDNA before and after bisulfite treatment should be reported (Fig 2), as the ambiguity with respect to size makes it very difficult to follow the comparison with the PCR fragment sizes used and specified in Fig 3 and the fragmentation of gDNA (300-1200 bp) (also Fig 3); the aforesaid clarification would help the discussion throughout the manuscript.

Response

The size of the cirDNA following bisulphite treatment cannot be determined from the agarose gel, because there are no ssDNA molecular weight markers available for comparison. However, we infer from the discrete nature of the bis-treated DNA bands that no fragmentation took place, since fragmentation is likely to have generated random sized pieces that migrate as a smear, rather than a discrete band. Lines 335-337 have now been amended to clarify this point. 

Comment

What is the nature of the “short DNA fragments” which are not found (line 67)? Do the authors mean the absence of fragmented bisulfite treated cirDNA which are smaller than the visualised sizes? It is important to accurately describe those sizes in the Results section, as “short” is relative and too vague in this context.

Response

We thank the reviewer for pointing this out, and have now amended the relevant section of the abstract to specify the size range of the fragments that were previously described as ‘short’, and as well as the discussion section (lines 421, 432 and 435-436).

Reviewer 2

Major points

Comment

1. Test system: The authors used Human Genomic DNA from human blood (buffy coat) as a high molecular weight (>50kb according to the manufacturer) reference DNA sample. Such HMW DNA is usually not present in clinically relevant samples. Accordingly, the test system is of limited use. This should be critically discussed in the manuscript.

Response

We agree with reviewer 2 that high molecular weight DNA, such as is shown in Figure 1 A and B is only rarely present in the circulating DNA fraction (e.g. Jahr et al., 2001), and that circulating DNA is generally low molecular weight. Data relevant to the low molecular weight size range of circulating DNA is presented in all of the figures following Figure 1B. 

In our manuscript, the results with the high molecular weight DNA serve several functions. Most importantly, they provide a positive control for DNA fragmentation in the bisulphite conversion protocols we carried out, showing that the low molecular weight DNA didn’t remain intact simply because the protocol was not applied correctly. 

Secondly, they allowed a comparison of the conversion efficiency of the two bisulphite kits tested. This could not be done on cirDNA, because the quantity that can readily be obtained from plasma is too low for analysis with the MGMT Pyro kit used to measure efficiency. 

Finally, a lane with control high molecular weight DNA is included in all gels that show low molecular weight DNA results, firstly, for reference, and secondly, to aid us in interpretation of results in case any unexpected HMW bands appeared in other lanes. 

In response to the reviewer comment, we have modified the discussion to include a sentence of the limitations of using high molecular weight DNA in this context (lines 383-386).

Comment

2. Novelty: Holmes et al. (PLoS One. 2014; 9(4): e93933.) have previously published that converting the same high molecular weight DNA using the same two Qiagen kits leads to >97% conversion; higher yields with the EpiTect Fast kit, and similar fragmentation patterns. Some of the results presented by Werner et al. are in perfect concordance with the results shown by Holmes et al.. Hence, the study by Holmes at al. should be appropriately included into the introduction and discussion of the paper.

Response

We have now included a reference to Holmes et al, 2014 when presenting the results (lines 272-275, 294-296 and 305-308). We note that Holmes et al only examined high molecular weight genomic DNA input, and all our data regarding cirDNA is novel. 

Comment

3. Influence of extraction method on fragmentation: Both investigated kits use silica membrane spin column-based DNA purification after bisulfite conversion. This purification might have an influence on the fragmentation of HMW DNA irrespective of the bisulfite conversion. The authors should perform a control reaction in which they omit the bisulfite reagent and replace it with water in order to test the influence of the extraction method on the fragmentation.

Response

The suggestion that interaction with the spin column might contribute to fragmentation of the DNA is interesting, although we note that spin column purification of genomic DNA from cell line and tissue samples does not generally result in a fragmented sample. We agree that isolating steps of the bisulphite conversion protocol to identify the one that is responsible for DNA degradation is worthwhile, but unfortunately beyond the scope of our current study. 

Comment

4. Sequence-specific fragmentation: The analyzed only two loci (MGMT and GSTP1). Fragmentation of DNA due to bisulfite treatment might be sequence-specific. Do AT-rich sequences show differences regarding fragmentation as compared to GC-rich sequences which are obviously much more affected by the bisulfite reaction? This matter should be discussed.

Response

We have now included a discussion of the influence of DNA sequence on bisulphite induced fragmentation (line 296- 299). 

5. In Figure 3A the authors showed that two PCR product showed the same band on the agarose gel before and after bisulfite conversion. The authors concluded that the fragmentation of the PCR product due to the bisulfite conversion is limited due to its small size and that the low concentration of the bisulfite converted PCR product is caused by the impaired visualization of ssDNA on an agarose gel. The results do not entirely support the conclusion. Since the authors used only 1% agarose gels, fragmented PCR products might not be visible. Other methods (e.g. 3.5% agarose gel, bioanalyzer, polyacrylamide gel) would be much better suited to visualize fragmented DNA. 

Response

We agree with Reviewer 2 that a high concentration (e.g. 3.5%) agarose gel would have been more appropriate to visualise the bis-treated DNA in Figure 2. We did initially run the bis treated cirDNA samples on 3% agarose gels, but weren’t able to see the DNA (data not shown), in retrospect, most likely due to a combination of insufficient input and running the gel for too long. We eventually turned to low (1%) concentration gels in an attempt to decrease the resolution of low molecular weight fragments, thereby compressing the anticipated smear of fragmented cirDNA and making it more readily visible. The finding that the bis-treated cirDNA was a discrete band was unanticipated. Repeating the experiment with a higher concentration gel would not substantially change our main finding that short DNA fragments, including cirDNA, undergo less degradation during bisulfite treatment. As the experiment consumes well over 100 mL of donor blood, we feel that running the same type of sample on a higher concentration gel is not justified. 

Comment

Secondly, the authors used PCR product as a surrogate DNA sample representing short DNA fragments. Even though the size is comparable to apoptotic DNA in plasma, PCR products might show a different behavior during the bisulfite conversion since the concentration of identical DNA sequences is much higher leading to a lower denaturation efficacy. It would be more convincing if the authors had compared the fragmentation of a large PCR product (~2kb) with their short ones (~200bp).

Response

The reviewer is suggesting that short PCR fragments and short cirDNA fragments resist bis fragmentation via two different mechanisms – high self complementarity and short length respectively. We agree that PCR products might show a lower denaturation efficiency due to a high concentration of complementary sequence, however, the denaturation conditions are very stringent (95°C), and sufficient to denature even high molecular weight DNA as is evidenced by the near-complete efficiency of unmethylated cytosine conversion in gDNA. Given that the PCR products are also completely denatured during the bis conversion, a more parsimonious explanation is that the stability of both PCR products and cirDNA is due to short length. We do acknowledge that self-complementarity might contribute to PCR product stability, but testing this experimentally is beyond the scope of the present study. 

Minor Points

Comment

1. Most scientist in the field use the spelling “bisulfite” instead of “bisulphite”. Using the more common spelling would increase the visibility of the article.

Response

We have changed the spelling of “bisulphite” to “bisulfite” throughout the manuscript.

Comment

2. Since the nature of the bisulfite conversion is a major part of this study, its features should be described more accurately. It is incorrect that methylated cytosine are unreactive. Their conversion occurs at much lower rates compared to cytosines. It is not entirely correct that DNA degradation is caused by the high temperature alone. DNA degradation is mainly caused by depurination and depyrimidation leading to abasic sites, followed by DNA strand breaks due to N-glycoside bond cleavage. This depurination and depyrimidation is mainly due to the low pH needed for the bisulfite conversion. A high temperature additionally increases this problem.

Response

We have amended the manuscript to include additional information on bisulphite conversion (line 91- 97).

Comment

3. Fig. 3: pb -> bp

Response

Figure 3 has now been amended to fix the error. 

Comment

4. Please use italicized gene names (IDH1, SNAI1, GSTP1).

Response

Gene names have now been italicised throughout the manuscript. 

Comment

5. The first paragraph extensively discusses features of gel electrophoresis regarding the visualization of small DNA fragments and in particular bisulfite converted small DNA fragments. Firstly, the scope of the study was not to describe the analytical performance of agarose gel electrophoresis, a method from the 1960s. 

Response

While agarose gel electrophoresis has been in use for a long time, its application to bisulphite treated circulating DNA, which is single stranded, is entirely novel. Our discussion of the performance of the technique relates almost entirely to single stranded DNA and is relevant to the new results presented in the paper. 

Comment 

Secondly, it is well known to people skilled in the art, that DNA disappears form an agarose gel after prolonged incubation, mainly due to diffusion. 

Response

We entirely agree that DNA bands on a gel decrease in intensity after prolonged electrophoresis, due to a mixture of diffusion and migrating beyond the gel stain. Our observation is that this effect much more pronounced with single stranded than with double stranded DNA. 

We draw the reviewers attention to the following feature of Figure 2 – the dinucleosome fragment from the 20 mL bis cirDNA has a stronger band than the 200 and 300 base pair molecular weight markers after 15 minutes of electrophoresis, but is almost invisible after 40 minutes of gel electrophoresis, while the 200 and 300 base pair molecular weight marker bands remain clear. 

To support this observation we have quantitated the relative intensities of the control and bis-treated PCR products shown in Figure 3 after 15 minutes and 40 minutes of electrophoresis. Our data, now included in the manuscript as supplementary Figure 1, clearly show that there is a much greater difference between the bis treated and the control PCR products after 40 minutes than after 15 minutes. Specifically, the control product is around 2-5 times more intense after 15 minutes, but around 15 times more intense after 40 minutes of electrophoresis. Hence, we conclude that while both single stranded and double stranded DNA are lost from agarose gels in the course of electrophoresis, the single stranded DNA is lost much more rapidly. 

Comment 

Thirdly, the authors used 1% agarose gel, a concentration that is not suited for small DNA fragments. In conclusion, the disappearance of small DNA fragments from the agarose gel after extended incubation time is most likely due to diffusion that is facilitated by the use of a too low concentrated agarose gel and prolonged incubation times. This paragraph of the discussion can be removed. In addition, Figures 2B and 3B can be removed since they contain basic knowledge that is already available to everyone in the field.

Response

As described above, the 1% gel was used in an attempt to compress the smear of fragmented low molecular weight DNA that we anticipated after bis-treatment of cirDNA. We have addressed the disappearance of the small DNA bands in our response to the previous comment, and with Supplementary Figure 1. Figures 2 B and 3B illustrate the more rapid loss of single stranded DNA, for both cirDNA and PCR products, which accounts for bisulphite treated cirDNA never having been visualised on a gel until now. As such, these figures are important to the study.

---

## [Decision Letter · Decision Letter 1]

19 Sep 2019

PONE-D-19-20341R1

Circulating cell-free DNA from plasma undergoes less fragmentation during bisulphite treatment than genomic DNA due to low molecular weight

PLOS ONE

Dear Dr Warton,

Thank you for submitting your manuscript to PLOS ONE. After careful consideration, we feel that it has merit but does not fully meet PLOS ONE’s publication criteria as it currently stands. Therefore, we invite you to submit a revised version of the manuscript that addresses the points raised during the review process.

One of the reviewers still has a few concerns regarding the manuscript that need further clarification, please see reviewer #3 comments.

We would appreciate receiving your revised manuscript by Nov 03 2019 11:59PM. To enhance the reproducibility of your results, we recommend that if applicable you deposit your laboratory protocols in protocols.io, where a protocol can be assigned its own identifier (DOI) such that it can be cited independently in the future. For instructions see: http://journals.plos.org/plosone/s/submission-guidelines#loc-laboratory-protocols

We look forward to receiving your revised manuscript.

Kind regards,

Baochuan Lin, Ph.D.

Academic Editor

PLOS ONE

Reviewers' comments:

Reviewer's Responses to Questions

**Comments to the Author**

1. If the authors have adequately addressed your comments raised in a previous round of review and you feel that this manuscript is now acceptable for publication, you may indicate that here to bypass the “Comments to the Author” section, enter your conflict of interest statement in the “Confidential to Editor” section, and submit your "Accept" recommendation.

Reviewer #2: All comments have been addressed

Reviewer #3: (No Response)

2. Is the manuscript technically sound, and do the data support the conclusions?

Reviewer #2: (No Response)

Reviewer #3: No

3. Has the statistical analysis been performed appropriately and rigorously? 

Reviewer #2: (No Response)

Reviewer #3: N/A

4. Have the authors made all data underlying the findings in their manuscript fully available?

Reviewer #2: (No Response)

Reviewer #3: Yes

5. Is the manuscript presented in an intelligible fashion and written in standard English?

Reviewer #2: (No Response)

Reviewer #3: Yes

6. Review Comments to the Author

Reviewer #2: (No Response)

Reviewer #3: There are several major concerns about this study. 1. The clinical meaning is not clear. How the community can use the findings? 2. Circulating DNA from patients with cancer or other severe diseases could be very different from healthy individuals. 3. The small sample size used can not describe sufficiently inter-individual variation.

7. PLOS authors have the option to publish the peer review history of their article (what does this mean?). If published, this will include your full peer review and any attached files.

Reviewer #2: No

Reviewer #3: No

---

## [Author Response · Author response to Decision Letter 1]

29 Sep 2019

We thank Reviewer 3 for the feedback on our manuscript. Specific responses to the comments are listed below.

1. The clinical meaning is not clear. How the community can use the findings? 

As described in the introduction, DNA methylation is a target of clinical assay development, particularly in the field of cancer diagnosis and monitoring. We are reporting a technical observation regarding the difference in fragmentation susceptibility between long and short DNA during methylation analysis protocols. In response to the reviewers comment, we have modified the introduction to clarify that our results apply to cancer detection tests and liquid biopsies targeting methylation of the low molecular weight fraction of cirDNA (line 104). This is the DNA size range that our results pertain to. We also note that our results were contrary to expectations (i.e. we expected bisulfite treatments to fragment cirDNA), and as such expand the understanding of the method for the research community developing PCR assays that target methylated cirDNA. 

2. Circulating DNA from patients with cancer or other severe diseases could be very different from healthy individuals. 

Reviewer 3 states correctly the circulating DNA size distribution might vary in cancer and other disease states. We have modified our discussion to emphasize that our results apply to DNA in the size range of ~170 bases, and cannot be extrapolated to disease states that change the size distribution of cirDNA to include high molecular weight fragments (lines 331-333). However, short DNA fragments are expected to behave in the same way, regardless of the additional presence of long fragments (please see comment below).

3. The small sample size used can not describe sufficiently inter-individual variation.

We are not aiming to describe individual variation in DNA susceptibility to fragmentation, rather, we aim to report at a mechanistic level, how a commonly used molecular biology technique (bisulfite conversion) affects DNA fragments in the size range of ~170 bases, including cirDNA. We have shown that this applies to DNA from two very different sources, that is cirDNA from a biological source (plasma) and a synthetic piece of DNA (i.e. PCR product). There are major differences between these two types of DNA e.g.: PCR product has uniform complementary sequence while cirDNA has diverse sequences; PCR product and cirDNA are likely to have different purification carry-over contamination, yet both types of DNA are affected by bisulfite treatment in the same way, as both are short. This suggests that inter-individual variation cirDNA other than size is not likely to change the effect.

Having said that, it would be interesting to examine inter-individual variation, however this is beyond the scope of this study. Since the effect we described is based on DNA length, it would be expected that cirDNA samples from individuals with longer DNA will undergo more fragmentation. However, this does not affect our conclusion that short DNA fragments are not fragmented during bisulphite treatment.

---

## [Editor Report · Decision Letter 2]

11 Oct 2019

Circulating cell-free DNA from plasma undergoes less fragmentation during bisulphite treatment than genomic DNA due to low molecular weight

PONE-D-19-20341R2

Dear Dr. Warton,

We are pleased to inform you that your manuscript has been judged scientifically suitable for publication and will be formally accepted for publication once it complies with all outstanding technical requirements.

With kind regards,

Baochuan Lin, Ph.D.

Academic Editor

PLOS ONE
---

## [Editor Report · Acceptance letter]

17 Oct 2019

PONE-D-19-20341R2 

Circulating cell-free DNA from plasma undergoes less fragmentation during bisulphite treatment than genomic DNA due to low molecular weight 

Dear Dr. Warton:

I am pleased to inform you that your manuscript has been deemed suitable for publication in PLOS ONE. Congratulations! Your manuscript is now with our production department. 

With kind regards,

on behalf of

Dr. Baochuan Lin 

Academic Editor

PLOS ONE